# Experimental Vertical Transmission of Chikungunya Virus by Brazilian and Florida *Aedes Albopictus* Populations

**DOI:** 10.3390/v11040353

**Published:** 2019-04-17

**Authors:** Nildimar Alves Honório, Keenan Wiggins, Bradley Eastmond, Daniel Cardoso Portela Câmara, Barry W. Alto

**Affiliations:** 1Laboratório de Mosquitos Transmissores de Hematozoários, Instituto Oswaldo Cruz, Fundação Oswaldo Cruz, 21040-360 Rio de Janeiro, Brasil; honorio@ioc.fiocruz.br (N.A.H.); dcpchamber@gmail.com (D.C.P.C.); 2Núcleo Operacional Sentinela de Mosquitos Vetores-Nosmove/Fiocruz, Fundação Oswaldo Cruz, 21040-360 Rio de Janeiro, Brasil; 3University of Florida, IFAS, Department of Entomology and Nematology, Florida Medical Entomology Laboratory, Vero Beach, FL 32962, USA; keenan.wiggins@gmail.com (K.W.); b-eastmond@hotmail.com (B.E.)

**Keywords:** *Aedes albopictus*, Chikungunya fever, vertical transmission, male role, inter-epidemic

## Abstract

Chikungunya virus (CHIKV) is a vector-borne alphavirus transmitted by the bites of mosquitoes, specifically infected, female mosquitoes of the invasive *Aedes* species. In nature, CHIKV can be maintained by vertical transmission, a phenomenon that relates to the transfer of CHIKV from the infected parent to their offspring within the ovary or during oviposition. In the present study, we conducted laboratory experiments to determine vertical transmission with *Ae. albopictus* populations from Brazil and Florida. Parental *Ae. albopictus* females were orally infected with the emergent Asian genotype of CHIKV in the first gonotrophic cycle (infectious blood meal) and tested for vertical transmission following the second (non-infectious blood meal) gonotrophic cycle. CHIKV infection and CHIKV viral titer in parental females were significantly related to population origin, with Brazilian *Ae. albopictus* showing higher viral dissemination and viral titer than the Florida population. Experimental vertical transmission of CHIKV was documented in one pool of female and four pools of male *Ae. albopictus* from Brazil (minimum infection rate, MIR, of 0.76% and 2.86%, respectively, for females and males). For the Florida population of *Ae. albopictus*, only one pool of males was positive for CHIKV infection, with an MIR of 1.06%. Our results demonstrate that *Ae. albopictus* populations from Brazil and Florida show heterogeneous CHIKV dissemination and vertical transmission, which may contribute to the epidemiology of CHIKV and may be particularly relevant to virus survival during inter-epidemic periods.

## 1. Introduction

Mosquito-borne arboviruses like dengue (DENV), Zika (ZIKV), and Chikungunya (CHIKV) are important public health challenges in tropical and subtropical regions [1,2]. Chikungunya virus (family *Togaviridae*, genus *Alphavirus*) is a vector-borne pathogen that originated in Africa and is maintained in two transmission cycles. The sylvatic cycle is between non-human primates and arboreal *Aedes* mosquitoes, while the urban cycle is between humans and invasive *Aedes*, where *Aedes aegypti* (L.) and *Aedes albopictus* (Skuse) are considered the main vectors [3,4,5]. Autochthonous cases of CHIKV has been documented in 112 countries, of which 47 countries or territories are in the Americas [6]. During 2018, Brazil has reported 87,687 probable cases [7], and 90 travel-associated cases were reported in the United States [8].

*Aedes albopictus* and *Ae. aegypti* are considered important invasive mosquitoes, successfully invading and colonizing human-dominated environments worldwide [9], and both species are widely distributed in the Americas [10]. The former species shows an eclectic feeding behavior, preferentially feeding and resting in the peridomicile, and it is more common in vegetated and urban/urban forest transition habitats, especially where it is sympatric with *Ae. aegypti* [9,11,12,13]. The later species is highly anthropophilic and exhibits endophilic behavior, feeding, and resting inside human dwellings, and it is mostly associated with high human density [11,14].

Both *Aedes* species from Brazil and Florida are considered competent vectors for CHIKV under experimental conditions [15,16,17]. So far, few reports have demonstrated natural infection of *Ae. aegypti* and *Ae. albopictus* by CHIKV in the Americas, especially Brazil [18]. One of the mechanisms by which arboviruses are maintained in nature is called vertical transmission, which can be defined as the direct transference of an arbovirus from one generation to the next [19,20], more specifically as the transmission of an arbovirus from an infected female mosquito to its offspring, regardless of the underlying mechanism [21]. Vertical transmission is considered a route of transmission that might allow for persistence of the virus during adverse environmental periods (e.g., dry and cold seasons, severe droughts), where adult mosquito proliferation and indirect transmission of a pathogen from arthropod vector to vertebrate hosts is not favorable, which thus allows for direct transmission (female parent to offspring). Because *Ae. aegypti* and *Ae. albopictus* eggs are desiccation-resistant, it is hypothesized that this trait could promote arbovirus survival throughout the year and between transmission cycles, playing an important role in endemic maintenance [19,22,23].

Although vertical transmission of CHIKV has been detected under natural conditions in field populations of *Ae. aegypti* and *Ae. albopictus* [24,25,26,27], and has been demonstrated under experimental conditions [28,29], there are conflicting reports in the literature [30,31,32]. In this paper, we determined experimentally for the first time the vertical transmission of the emergent Asian genotype of CHIKV in Brazilian and Florida populations of *Ae. albopictus*.

## 2. Materials and Methods

### 2.1. Ethics Statement

Chikungunya virus (Asian lineage, GenBank accession: KJ451624) used in the study was isolated from the serum of an infected human in the British Virgin Islands in 2013 by other investigators. Subsequently, this isolate was archived with the Centers for Disease Control and Prevention. We requested an isolate of this virus for use in this study, and so the sample was already present in an already existing collection (Centers for Disease Control and Prevention, Arboviral Diseases Branch). The virus sample was anonymized, and Institutional Review Board approval was not needed for receipt and use of the sample in this study. No entomological gathering was done on private land or in private residence for this study.

### 2.2. Mosquito Populations

The *Ae. albopictus* strain used in this experiment were F2 from Rio de Janeiro (RJ), Brazil, and Okeechobee (OK), Florida, United States, respectively. All the gathering of entomological samples was done on public land. In Rio de Janeiro, *Ae. albopictus* eggs were obtained at the Oswaldo Cruz Foundation campus in March 2015 from 50 oviposition traps. In Okeechobee, immatures of sympatric *Ae. albopictus* were collected from tires in October 2015. Details on population characteristics and collection methods can be found elsewhere [15].

### 2.3. Mosquito Rearing

To stimulate *Ae. albopictus* egg hatching, oviposition papers were added to pans with 1 L of tap water and 0.2 g of 1:1 brewer’s yeast and lactalbumin. After hatching, first instar larvae were redistributed to 100 larvae per pan. Supplementary larval food resources (0.2 g brewer’s yeast and lactalbumin) were added to each pan every other day for approximately 10 days until individuals pupated or died. Upon pupation, pupae were collected daily and placed in vials with a cotton seal until eclosion into 0.3 m^3^ adult cages. After emergence, female and male mosquitoes were held together within the cages for eleven days to allow for mating. During this time, around 50 *Ae. albopictus* females were separated per box and were provided with 10% sucrose solution and water at all times. Bloodmeals were provided at 7–8 (infectious blood) and 13 days (uninfected blood) after emergence. The mosquitoes were held in a climate-controlled room at 26–28 °C and a photoperiod of 14:10 hours light:dark.

### 2.4. Virus and Mosquito Oral Infection

We used an isolated strain of CHIKV (Asian lineage, GenBank accession: KJ451624) from the serum of an infected human in the British Virgin Islands in 2013. The CHIKV isolate was passaged twice in culture using African green monkey (Vero) cells, and viral titer was determined in 6-well plates seeded with Vero cells by plaque assay using a modified procedure elsewhere [33]. For virus suspension, monolayers of Vero cells were inoculated with dilute stock CHIKV at a multiplicity of infection of 0.1, followed by a one-hour incubation at 37 °C and 5% carbon dioxide atmosphere. After the inoculation procedure, each flask received 24 mL media (M199 medium supplemented with 10% fetal bovine serum, penicillin/streptomycin and mycostatin) and was left to incubate for an additional 47 h. The *Ae. albopictus* females were provided with two serial blood meals during the experiment (Figure 1). For the first blood meal infection, 7–8 day old females were offered CHIKV-infected, defibrinated bovine blood (Hemostat, Dixon, CA, USA) using an artificial feeding system with hog intestine membranes (Hemotek, Lancashire, United Kingdom). For the second gonotrophic cycle, 13 day old females fed on uninfected blood meals for 30 min. In both feeding trials, *Ae. albopictus* females were classified as successfully fed, partially fed, unfed, and dead. During the first and second gonotrophic cycles, *Ae. albopictus* females were kept in cages (around 50 individuals per cage). Samples of blood were taken from the virus–blood suspension at the time of feeding to determine the concentration of CHIKV ingested by the adult mosquitoes. The titer of infectious blood meals was 8.3 log_10_ plaque forming units (pfu)/mL. Fully engorged females were held in cylindrical cages (10 cm ht. × 10 cm top dia. × 7 cm bottom dia.) along with an oviposition substrate and maintained at a 14:10 hour light:dark photoperiod and 28 °C. Mosquitoes were stored at −80 °C after the transmission assay and later dissected to test the legs for the presence of CHIKV RNA by qRT-PCR. The sequence of primers (accession ID of transcript, KU365292.1) targeting a nonstructural polyprotein gene was as follows: forward, 5′-GTACGGAAGGTAAACTGGTATGG-3′; reverse, 5′-TCCACCTCCCACTCCTTAAT-3′. The probe sequence was: 5′-/56-FAM/TGCAGAACCCACCGAAAGGAAACT/3BHQ_1/-3′ (Integrated DNA Technologies, Coralville, IA, USA). Detection of CHIKV RNA in the legs of mosquitoes was regarded as a disseminated infection [15].

Female legs were triturated in 1.0 mL of media (GIBCO^®^ Media 199). RNA isolation on a 140 µL sample of mosquito legs homogenate was achieved using the QIAamp viral RNA mini kit (Qiagen, Valencia, CA, USA) and eluted in 50 µL of buffer according to the manufacturer’s protocol. Viral RNA was detected using the Superscript III One-Step qRT-PCR with Platinum^®^ Taq kit by Invitrogen (Invitrogen, Carlsbad, CA, USA) using methods described elsewhere. Quantitative RT-PCR was performed with the CFX96 Real-Time PCR Detection System (Bio-Rad Laboratories, Hercules, CA, USA) with the following program: 50 °C for 30 min, 94 °C for 2 min, 39 cycles at 94 °C for 10 s and 60 °C for 1 min, and 50 °C for 30 s.

### 2.5. Maintenance of Adults

Females maintained in boxes were held within incubators at 28 ± 0.5 °C on a daily photoperiod of 14:10 hour light:dark, during and after feeding on the infectious and uninfected bloodmeals. Females were allowed to lay eggs within 3–5 days during their first and second gonotrophic cycle. After the first oviposition, female mosquitoes were provided with an uninfected blood meal. Fully engorged mosquitoes were placed individually in 37-mL plastic tubes, which were covered with a screen and lined internally with moist germination paper as an oviposition site. A cotton ball soaked with 10% sucrose solution was placed on top of the screen of each tube and renewed daily. After females laid eggs during their second gonotrophic cycle, they were individually stored in microcentrifuge tubes at −80 °C. Eggs on germination papers were counted (except first oviposition) and held in an incubator at 85% relative humidity (RH) until the infection status of each female parent was determined.

### 2.6. Determination of Progeny Infectious Status

Using only egg batches from females with a disseminated infection in the second gonotrophic cycle, hatching was stimulated by immersing eggs and 0.1 g larval food in individual 0.47-L, water-filled, plastic containers. The progeny from each female parent were provided with larval food every other day until they developed to the adult stage; then, they were killed by freezing, pooled, and stored in microcentrifuge tubes at −80 °C as described above. The pools of whole adult mosquitoes that originated from the second gonotrophic cycle were then tested for the presence of CHIKV RNA using the methods previously described.

### 2.7. Statistical Analyses

Descriptive statistics were calculated for the different life stages and pools tested for CHIKV for the progeny from the second gonotrophic cycle of Brazilian (Rio de Janeiro) and US (Florida, Okeechobee) *Ae. albopictus*. Exploratory analyses involved boxplots, bar plots, and frequency tables. Analysis were done to analyze CHIKV viral dissemination (a dichotomous dependent variable) and viral titer in parental females and their progeny (two continuous dependent variables). As independent variables, we measured population origin (Rio de Janeiro and Florida) and vertical transmission (progeny infected or uninfected). We used a generalized linear model of the binomial distribution to model CHIKV viral dissemination in parental females. We used a one-way ANOVA to model viral titer in parental females and viral titer in their progeny. In order to attend to homoscedasticity and Gaussian distribution of the residuals of the ANOVA, we used the Box–Cox procedure to analyze the ANOVA results and transformed the dependent variables by power transformation. All analyses were done using R [34] and RStudio [35], with the library MASS [36].

## 3. Results

### 3.1. Parental Female Feeding and Oviposition

Of the 500 parental Brazilian and 501 Florida *Ae. albopictus* females reared for the experiment, 401 (80.2%) and 407 (81.24%) were successfully fed respectively at the end of the first gonotrophic cycle. The percentage of successfully fed parental females decreased to 56.36% and 58.23% in the second gonotrophic cycle, respectively, for the Brazilian and Florida populations of *Ae. albopictus*. Table 1 shows the descriptive statistics regarding the total amount of deposited eggs, pupae and emerged adults from the parental females in the second gonotrophic cycle of the experiment, and the total number of pools testing positive for CHIKV RNA. Of the 5299 deposited eggs by Brazilian *Ae. albopictus*, 53.18% successfully reached the pupae stage and 51.10% the adult stage, with a pupae mortality of 3.90%. Of the 3384 deposited eggs by Florida *Ae. albopictus*, 55.70% reached the pupae stage and 53.43% the adult stage, with a pupae mortality of 4.08%.

### 3.2. CHIKV Dissemination in Parental Females and Vertical Transmission

A total of 64.60% Brazilian *Ae. albopictus* and 44.30% Florida *Ae. albopictus* parental females were positive for disseminated infection at the end of the second gonotrophic cycle. Viral dissemination of parental females was significantly related to population origin, with the Brazilian *Ae. albopictus* showing higher viral titer than the US population (Mean ± SE, 90.12% ± 2.35% and 82.03% ± 3.41%, respectively) (Figure 2, Table 2). The viral titers of parental females were significantly related to population origin, with the Brazilian *Ae. albopictus* showing a higher viral titer than the US population (Log10 Mean ± SE, 4.06 ± 0.11 and 3.35 ± 0.16, respectively). (Figure 3, Table 2). We did not find a significant relationship between the viral dissemination of parental females and vertical transmission (due to non-convergence of the model). Also, we did not find a significant relationship between the viral titer of parental females and vertical transmission, and between the viral titer of the progeny and population origin or vertical transmission (Table 2). One pool of female and four pools of male *Ae. albopictus* from Brazil were positive for CHIKV infection (minimum infection rate, MIR, of 0.76% and 2.86%, respectively, for females and males). For the Florida population of *Ae. albopictus*, only one pool of males was positive for CHIKV infection, with an MIR of 1.06% (Table 1).

## 4. Discussion

During 2004 and the following ten years, CHIKV was responsible for one of the most widespread and unprecedented emergences seen in the history of vector-borne diseases, spreading through Africa, Asia, the Indian Ocean, and Europe [37]. In the Americas, CHIKV was first detected in the Caribbean island of Saint Martin in 2013, from where it rapidly spread to several other countries and territories in the region, reaching South and Central America soon after its detection [1,38]. In Brazil, two different CHIKV introductions resulted in the simultaneous circulation of two CHIKV lineages, the Asian and East/Central/South African (ECSA) genotypes [39,40]. In continental US, most CHIKV infections in humans have been primarily travel-associated (475 cases), with 12 locally-transmitted human cases reported in Florida in 2014 [41].

In the present study we tested experimentally the vertical transmission of the emergent Asian genotype of CHIKV in Brazilian and Florida populations of *Ae. albopictus*. We carried out experiments to determine the vertical transmission of CHIKV in low-passaged *Ae. albopictus* reared from infected females. In our results, viral dissemination of parental females was significantly related to population origin, with the Brazilian *Ae. albopictus* showing higher viral titer than the US population (Mean ± SE, 90.12% ± 2.35% and 82.03% ± 3.41%, respectively). Similar viral dissemination rates were obtained for different Brazilian and Florida populations of *Ae. aegypti* and *Ae. albopictus* at 2, 5, and 13 days post-infection for the emergent Asian genotype of CHIKV [15]. Our results also showed that all tested populations of both *Aedes* had a high proportion (>0.80) of individuals with a disseminated infection as early two days post-exposure. For *Ae*. *albopictus*, both American and Brazilian populations had similar dissemination rates on the second day (0.843 ± 0.065 and 0.827 ± 0.071). Transmission rates had a heterogeneous pattern, with Brazilian *Ae. albopictus* having the highest proportion of individuals with a successful infection (0.82 at two days postinfection).

In our results, the minimum infection rates obtained for adults was documented in one pool of female and four pools of male *Ae. albopictus* from Brazil (minimum infection rate, MIR, of 0.76% and 2.86% respectively for females and males). For the Florida population of *Ae. albopictus*, only one pool of males was positive for CHIKV infection, with an MIR of 1.06%. According to Adams and Boots [42], mathematical models showed that in endemic situations, increases in reproductive number, half-life, and persistence of the disease only become significant when vertical infection efficiency exceeds 20–30%. Although our results show an MIR below the level proposed by the authors, we believe that further laboratory and field investigations should be done to quantify vertical transmission under experimental and field situations, to further study the importance of this type of arbovirus transmission in the epidemiology of CHIKV, especially in places where *Ae. albopictus* might act as the main or secondary vector. Additionally, further studies are needed to measure the filial infection rate (percentage of the infected female parent offspring that become infected). Vertically infected mosquitoes are likely strong contributors to transmission because they are, presumably, capable of transmission upon first bite. In contrast, horizontally infected mosquitoes require the ingestion of one or more infected blood meals to become infected, followed by a time lag associated with the extrinsic incubation period. Taken together, vertically infected mosquitoes are “infectious” for a greater proportion of their adult lifespan than horizontally infected individuals and a greater contributor to vectorial capacity (i.e., the extrinsic incubation period of adults is negligible). Our method of calculating MIR likely underestimates the relative importance of vertical transmission of *Ae. albopictus* to the transmission of CHIKV [43].

Our results show that CHIKV infection was evident in mosquito progeny from the second gonotrophic cycle, a result that contrasts with those found in another study [28]. Our results are similar to those found by Bellini et al. [44]. These authors performed a series of experimental infections in *Ae. albopictus* females with different strains of CHIKV derived from the 2007 outbreak in Italy to measure, among other variables, vertical transmission to progeny. The authors did not find infected individuals from the first gonotrophic cycle, but found two infected males and one infected female from the second gonotrophic cycle [44]. The authors hypothesize that the late infection progeny (at the second gonotrophic cycle) might be due to a longer time needed for the CHIKV to infect the female’s ovarioles. However, in the study done by Chompoosri et al. [28], vertical transmission of CHIKV was detected as early as the first gonotrophic cycle. Also, we detected CHIKV infection in one pool of females and four pools of male *Ae. albopictus* from Brazil and one pool of males from Florida, showing a heterogeneous aspect of vertical transmission in different populations in the Americas. The higher number of infected *Ae. albopictus* male pools may show that venereal transmission (i.e., the passage of viral particles carried in semen from vertically infected males to females during coition [19]) might be another contributor to the CHIKV maintenance in endemic areas.

In fact, natural vertical transmission of CHIKV has been reported in different parts of the world in the years since its emergence, especially during outbreaks and epidemics of this arbovirus. In Reunion Island, CHIKV was detected in two pools of *Ae. albopictus* from a total of 502 pools during a large entomological survey [24]. During the 2005–2006 outbreak of CHIKV in this territory, ~38% of the human population was affected, and *Ae. albopictus* was the responsible vector because of a single amino acid substitution (substitution of alanine by valine at position 226) in the E1 glycoprotein [45]. In India, CHIKV with the same mutation in the E1 glycoprotein was detected in adult *Ae. albopictus* reared from field-collected larvae during the 2009 outbreak in Kerala [46]. In Thailand, CHIKV was detected in both females and males of field-collected specimens of *Ae. aegypti* and *Ae. albopictus*, with the latter showing a higher infection rate than the former [26]. During a concurrent outbreak of DENV and CHIKV in Madagascar, *Ae. albopictus* was the only vector found in entomological surveys. Furthermore, adults reared from field-collected larvae were found to be positive for CHIKV infection [25]. Together, these results show that the impact of vertical transmission in CHIKV transmission dynamics needs to be further studied, especially in areas where *Ae. albopictus* is present and sylvatic cycles are absent.

Data on the vertical transmission of CHIKV under experimental conditions are scarce and have conflicting reports. An experiment done with *Ae. aegypti* and *Ae. albopictus* using intrathoracic injection showed that none of the progeny of both species were infected by CHIKV after three gonotrophic cycles [30]. In an experiment using *Ae. albopictus* originating from La Réunion that were orally infected with a blood meal containing 10^8^ pfu/mL of the CHIKV, the authors were not able to detect vertical transmission in two gonotrophic cycles [31]. Another experiment, done with *Ae. albopictus* and CHIKV from the 2007 Italy outbreak, revealed one female and two males positive for vertical transmission after two gonotrophic cycles [44]. In a series of experiments, Chompoosri et al. [28] tested vertical transmission of CHIKV in *Ae. aegypti* and *Ae. albopictus* on multiple progeny. The authors found that *Ae. aegypti* females were able to transmit CHIKV vertically to F5 progeny, while *Ae. albopictus* females transmitted to F6 progeny, showing that the latter species was more susceptible to vertical transmission of CHIKV than the former. Furthermore, the authors were able to detect CHIKV in *Ae. aegypti* and *Ae. albopictus* larvae and adults developed from eggs of the first gonotrophic cycle, while we were able to detect CHIKV only in the progeny from the second gonotrophic cycle in the present study [28]. Agarwal and coauthors [29] conducted the laboratory experiments to determine whether *Ae. aegypti* from India is capable of vertically transmitting a novel ECSA genotype of CHIKV. The authors showed that only the larvae and adults developed after subsequent noninfectious blood meal (third gonotrophic cycle) with high positivity (MIR 28.2 and 20.2, respectively).

Horizontal transmission between infected *Aedes* mosquitos and humans is the only method clearly linked to transmission of CHIKV. However, vertical transmission of arboviruses remains an important characteristic responsible for the maintenance of endemic/epidemic arbovirus transmission cycles, especially during unfavorable environmental conditions for transmission by adult mosquitoes [19,47]. Our study adds more evidence that vertical transmission might be an important characteristic in the epidemiology of CHIKV, demonstrating that *Ae. albopictus* mosquitoes from Brazil and Florida may contribute to a better understanding of the epidemiology of CHIKV through vertical transmission, which may be particularly relevant to virus survival during interepidemic periods.

## Figures and Tables

**Figure 1 viruses-11-00353-f001:**
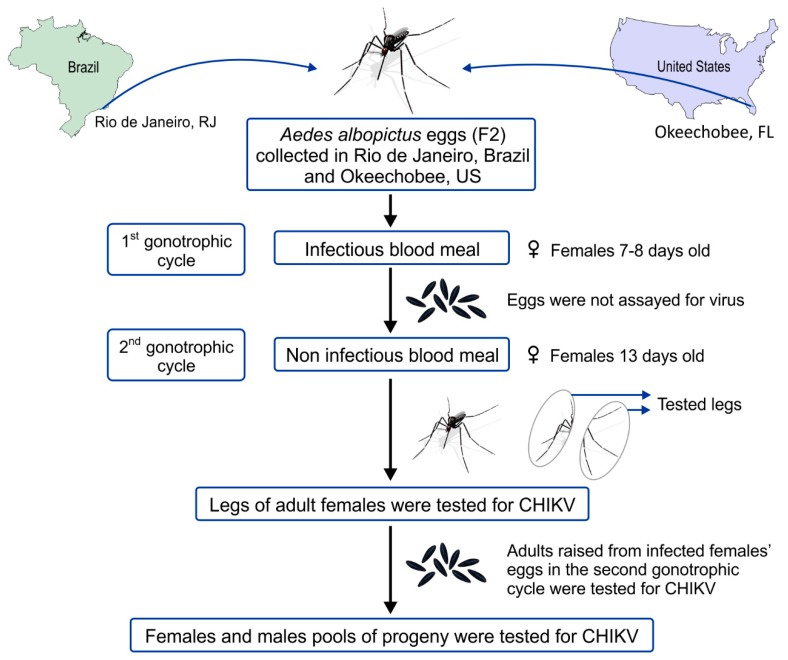
Experimental design of the laboratory vertical transmission of Chikungunya (CHIKV) by *Aedes albopictus* populations [29].

**Figure 2 viruses-11-00353-f002:**
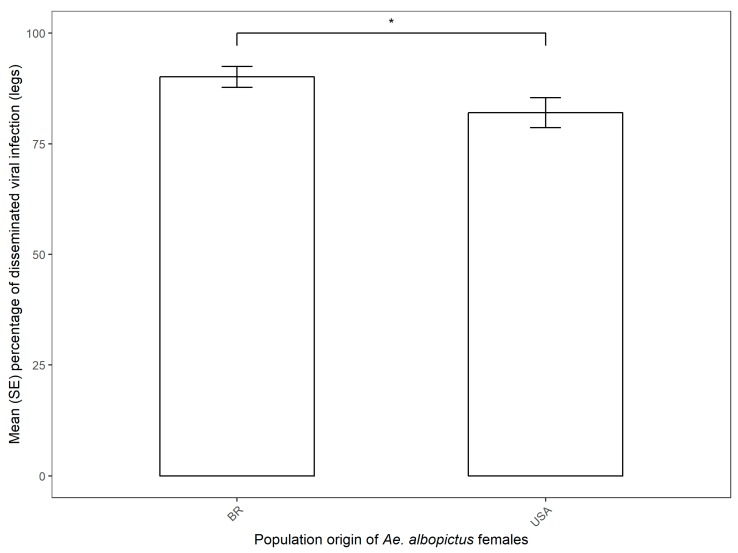
Mean (SE) dissemination rate of CHIKV in *Ae. albopictus* from Brazilian and Florida populations. Statistical analysis was done via a binomial generalized linear model, with a significant effect of population origin highlighted by the comparison bar (* *z* < 0.05, ** *p* < 0.01, *** *p* < 0.001).

**Figure 3 viruses-11-00353-f003:**
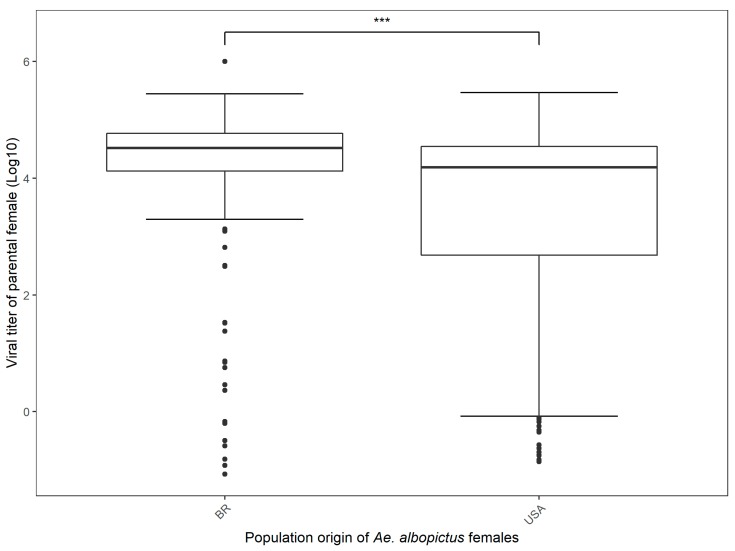
Boxplot of viral titer (log 10) of parental female in *Ae. albopictus* populations from Brazilian and Florida populations. Statistical analysis was done by a one-way ANOVA, with a significant effect of population origin highlighted by the comparison bar (* *p* < 0.05, ** *p* < 0.01, *** *p* < 0.001).

**Table 1 viruses-11-00353-t001:** Descriptive statistics for the different life stages and pools tested for CHIKV for the progeny from the second gonotrophic cycle of Brazilian (Rio de Janeiro) and US (Florida, Okeechobee) *Ae. albopictus*. Mosquito stages describe each of the stages that were analyzed. Descriptive statistics refers to the total number of each mosquito stage at the end of the second gonotrophic cycle, the mean (standard deviation) for each mosquito stage, and the minimum and maximum observed values for each stage. Adults were separated into two entries, one for females and another for males. Positive/total adult pools refers to the absolute number of pools that were tested positive for CHIKV vertical transmission infection and the total number of pools that were tested for each mosquito sex. Minimum infection rate stands for the number of positive pools/total number of tested mosquitos (stratified by sex) × 1000, as per Centers for Disease Control and Prevention (CDC) recommendations.

Mosquito Stages	Descriptive Statistics	*Ae. albopictus* Population	
		Rio de Janeiro/BR	Okeechobee/US
**Eggs**	**Total**	5299	3384
	**Mean (SD)**	23.14 (25.80)	26.23 (25.86)
	**Minimum–Maximum**	0–119	0–121
**Pupae**	**Total**	2818	1885
	**Mean (SD)**	20.13 (16.82)	20.71 (20.85)
	**Minimum-Maximum**	0–86	1–107
**Adult females**	**Total**	1309	866
	**Mean (SD)**	9.49 (8.06)	9.52 (9.75)
	**Minimum–Maximum**	0–43	0–46
	**Positive/Total Female Pools**	1/138	0/94
	**Minimum Infection Rate**	0.76	0
**Adult males**	**Total**	1399	942
	**Mean (SD)**	10.14 (8.67)	10.35 (10.76)
	**Minimum–Maximum**	0–38	0–60
	**Positive/Total Male Pools**	4/147	1/102
	**Minimum Infection Rate**	2.86	1.06

**Table 2 viruses-11-00353-t002:** Estimated effects for the following models: (1) Binomial generalized linear model for viral dissemination in parental females versus population origin; (2) binomial generalized linear model for viral dissemination on parental females versus vertical transmission; (3) one-way ANOVA for viral titer of the parental females versus population origin; (4) one-way ANOVA for viral titer of the parental females versus vertical transmission; (5) one-way ANOVA for viral titer of the progeny versus population origin; and (6) one-way ANOVA for viral titer of the progeny versus experimental vertical transmission. Entries in bold and italic indicate statistical significance (*p* < 0.05).

Response	Model	Effect	Estimate	Standard Error	95% CI	
					Lower Bound	Upper Bound
**Viral dissemination (parental females)**	#1	(Intercept)	9.125	0.263	5.625	15.904
		Population origin: USA	0.500	0.350	0.248	0.987
	#2	(Intercept)	6.312	0.122	5.007	8.067
		Vertical transmission: positive	2.376	1.566	NA	NA
**Viral titer (parental females)**	#3	(Intercept)	12.607	0.426	11.773	13.441
		Population origin: USA	−2.913	0.642	−4.172	−1.654
	#4	(Intercept)	11.332	0.334	10.678	11.987
		Vertical transmission: positive	−0.143	2.274	−4.599	4.314
**Viral titer (progeny)**	#5	(Intercept)	1.306	0.033	1.241	1.372
		Population origin: USA	−0.006	0.053	−0.111	0.099
	#6	(Intercept)	2.462	0.099	2.268	2.656
		Vertical transmission: positive	−0.580	0.886	−2.316	1.155

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
