# Peer review of "Experimental Vertical Transmission of Chikungunya Virus by Brazilian and Florida Aedes Albopictus Populations"

_viruses, 2019, doi:10.3390/v11040353_

Reviewer 1 Report

The study ‘VERTICAL TRANSMISSION OF CHIKUNGUNYA VIRUS BY BRAZILIAN AND FLORIDA AEDES ALBOPICTUS POPULATIONS’ uses valuable mosquito populations native to Brazil and Florida to investigate the experimental vertical transmission rate of CHIKV strain KJ451624 in these vectors. Investigating the potential of these vector species to transmit CHIKV vertically and potentially impact the spread or cause of outbreaks of this emerging virus is of interest. This is particularly true given that CHIKV is a major concern globally. The introduction and discussion of the manuscript are well referenced. The manuscript is well written with minor English errors that require attention. The Materials and Methods are described well and Figure 1 is an essential part of this. There are concerns regarding the presentation of results, which should be addressed by the authors.

Major revisions

Can the authors explain why a control mosquito population, lab strain of Ae. Albopictus, was not used to compare with field mosquitos?

The authors need to make clear in the manuscript that they are investigating “experimental” vertical transmission and not natural vertical transmission.

All figure and table legends need attention and more detail. 

Table 1 – “Stages/Pools” written twice? What is the mean (SD) of? What is the minimum-maximum of? What is Positive/Total pool? What is MIR? All this detail needs adding, at least to the legend, to improve clarity.

If differences in the data of figures is significant then this should be made clear in the figure by adding annotation. The method used for statistical analysis should also be added to the figure legend.

Figure 3 – What method was used to analyse virus titer? Units are not presented in figure. If differences are significant then add detail and analysis performed to legend.

Table – What does the numbering relate to in the legend? It doesn’t seem to relate to the table. “Bold” entries are not able to be made out in print offs. Another method of highlighting significance would be useful.

Minor revisions

English corrections throughout the manuscript: Line 40, Line 41, Line 146, Line 181, Line 188-191 (too long difficult to understand), Line: 202 (“on” makes it difficult to understand what is being tested), Line 220 (is “significantly” used appropriately to described statistical significance?), Line 226 (Font of Ae Albopictus)

Author Response

Dear Editor,

 MS Viruses-477811 titled “Vertical transmission of chikungunya virus by Brazilian and Florida Aedes albopictus populations”

 Thank you for the reviews of our manuscript. We really appreciate the referees for their thoughtful suggestions and comments. We have addressed the reviewer’s comments and made the necessary edits to the manuscripts.

We hope you find our comments / edits satisfactory and that the paper is suitable for publication.

 Yours sincerely,

Drs. Honório and Alto

Corresponding authors

Comments from Reviewers and Reply to the referee’s comments:

Reviewer #1:

The study “Vertical transmission of chikungunya virus by Brazilian and Florida Aedes albopictus populations” uses valuable mosquito populations native to Brazil and Florida to investigate the experimental vertical transmission rate of CHIKV strain KJ451624 in these vectors. Investigating the potential of these vector species to transmit CHIKV vertically and potentially impact the spread or cause of outbreaks of this emerging virus is of interest. This is particularly true given that CHIKV is a major concern globally. The introduction and discussion of the manuscript are well referenced. The manuscript is well written with minor English errors that require attention. The Materials and Methods are described well and Figure 1 is an essential part of this. There are concerns regarding the presentation of results, which should be addressed by the authors.

Answer: We thank reviewer #1 for providing a thorough review and helpful suggestions and comments.

Major revisions

1.    Can the authors explain why a control mosquito population, lab strain of Ae. albopictus, was not used to compare with field mosquitos?

Answer: We agree that it would have been useful to include one or more additional strains of Ae. albopictus to identify variation in infection and transmission measurements. However, our primary objective was to make use of a low passage mosquito and CHIKV virus more closely representative of natural populations.

These experiments are labor intensive and so the addition of another mosquito strain would necessitate smaller samples size. We viewed this as potentially problematic because we anticipated low rates of vertical transmission. Therefore we needed to maximize our sampling effort for only one mosquito strain for Florida and Brazil.

 2.    The authors need to make clear in the manuscript that they are investigating “experimental” vertical transmission and not natural vertical transmission.

 Answer: As suggested by the reviewer, we made clear in the manuscript that we are investigating “experimental” vertical transmission.

 We have made this point clearer in the title, abstract, and introduction sections.

 3.    All figure and table legends need attention and more detail.

 Answer: We thank the reviewer for the valuable comments regarding results presentation. All figures and tables have been changed accordingly. Changes are specified below each comment.

4.    Table 1 – “Stages/Pools” written twice? What is the mean (SD) of? What is the minimum-maximum of? What is positive/Total pool? What is MIR? All this detail needs adding, at least to the legend, to improve clarity.

 Answer: We would like to thank the reviewer for bringing to our attention that Table 1 was not sufficiently detailed, and that “Stages/Pools” was written twice. Table 1 has been changed, so that descriptive statistics are more presentable to the reader. We added a column to the left of the table, indicating what is the mosquito stage (eggs, pupae, adult female and adult male) that the descriptive statistics are referred to. We also included the following description of how each descriptive statistics was calculated after the table title: “Mosquito stages describe each of the stages that were analyzed. Descriptive statistics refers to the total number of each mosquito stage at the end of the second gonotrophic cycle, the mean (standard deviation) for each mosquito stage and the minimum and maximum observed values for each stage. Adults were separated into two entries, one for females and another for males. Positive/Total adult pools refers to the absolute number of pools that were tested positive for CHIKV vertical transmission infection and the total number of pools that were tested for each mosquito sex. Minimum Infection Rate stands for the number of positive pools / total number of tested mosquitos (stratified by sex) x 1,000, as per CDC recommendations.”

5.    If differences in the data of figures is significant then this should be add clear in the figure by adding annotation. The method used for statistical analysis should also be added to the figure legend.

 Answer: Thanks for bringing this to our attention. We added a significance bar indicating comparison between population origin and an asterisk in both Figures 2 and 3 to make clear that statistical comparisons were significant under an alpha of 0.05. We also added the following explanation after the title in Figure 2 to make it more understandable: “Statistical analysis was done via a binomial generalized linear model, with a significant effect of population origin highlighted by the comparison bar (* = p < 0.05, ** = p < 0.01, *** = p < 0.001)”. Also, we added the following after Figure 3 title: “Statistical analysis was done by a one-way ANOVA, with a significant effect of population origin highlighted by the comparison bar (* = p < 0.05, ** = p < 0.01, *** = p < 0.001).”

6.    Figure 3 – What method was used to analyse virus titer? Units are not presented in figure. If differences are significant then add detail and analysis performed to legend.

 Answer: To make the manuscript clearer for the reader, we expanded the statistical analysis in the methodology section. The following text was added to the mentioned section: “Analysis were done to analyze CHIKV viral dissemination (a dichotomous dependent variable) and viral titer in parental females and their progeny (two continuous dependent variables). As independent variables, we measured population origin (Rio de Janeiro and Florida) and vertical transmission (progeny infected or uninfected). We used a generalized linear model of the binomial distribution to model CHIKV viral dissemination in parental females. We used a one-way ANOVA to model viral titer in parental females and viral titer in their progeny”. As said in the comment above, we also added the following explanation after the title in Figure 2 to make it more understandable: “Statistical analysis was done via a binomial generalized linear model, with a significant effect of population origin highlighted by the comparison bar (* = p < 0.05, ** = p < 0.01, *** = p < 0.001).”

7.    Table – What does the numbering relate to in the legend? It doesn’t seem to relate to the table. “Bold” entries are not able to be add out in print offs. Another method of highlighting significance would be useful.

 Answer: We would like to thank the reviewer for bringing to our attention that only using bold entries was not clear. In order to make significant effects more apparent, we used bold and italic fonts. Also, we expanded the models description with information on what were the analysis used (binomial GLM and one-way ANOVA) and we added a column entitled “Model” so that the numbering can be easily related to each statistical model.

 Minor revision

 English corrections throughout the manuscript: Line 40, Line 41, Line 146, Line 181, Line 188-191 (too long difficult to understand), Line: 202 (“on” makes it difficult to understand what is being tested), Line 220 (is “significantly” used appropriate to described statistical significance?), Line 226 (Font of Ae. albopictus)

Answer: We have rephrased these sections as requested.  For line 220 (original submission), significance is appropriated described. 

Reviewer 2 Report

With great interest, I read the paper of Honorio et al. submitted for publication in Viruses. The authors provide data that can help to interpreted the possible role of vertical transmission in epidemiology of arboviruses (here chikungunya virus) in the field. The fact that they found heterogeneous CIKV dissemination and vertical; transmissions . The work has been executed with great care. However I would like the authors to address the following, rather editorial issues.

Scientific comment: It is not clear to me from the Material Methods whether the females held individually or in cages (how many). It would be interesting to know whether the variation was due to variation between females (did e.g. did for example only one infected female infected eggs). Please elaborate.

Editorial comments:

Page 1, line 18: Please add “mosquito” before infected parent as it is unclear during reading whether you mean the vector or humans.

Page 3 line 92, spacing between 0.3m3 is missing

Page 3 line 93: omit extra “t” from mating

Page 4 Figure 1: I would redraw this diagram as it rather confusing as the text boxes do not contain comparable content. It is not clarifying what is done and what is tested. Further:

I would add that the original Aedes albopictus were F2 from mosquito eggs collected from the two countries.

Why are there circles around the second mosquito in the diagram? Because legs were tested? Confusing

Since you did not test virus in eggs, but in adults raised from eggs from infected females, I would rephrase the text next to the second and latest batch eggs in the diagram

Page 5:  line 173-175. The numbers in the text do not agree with those shown in the table. In addition, the Table 1 is confusing to me. Please check the headings of the columns. Further Total and mean of what (pools?).  Minimum-maximum of what (per female, per pool, per test, per cage???).

Page 6: Figure 2. The content of the graph could be easily reflected in a table or joint in figure 3.

Page 8: Line 224. It is a bit odd to refer to “these authors when the majority is the same as the authors of this paper. Please rephrase.

Author Response

Dear Editor,

MS Viruses-477811 titled “Vertical transmission of chikungunya virus by Brazilian and Florida Aedes albopictus populations”

Thank you for the reviews of our manuscript. We really appreciate the referees for their thoughtful suggestions and comments. We have addressed the reviewer’s comments and made the necessary edits to the manuscripts.

We hope you find our comments / edits satisfactory and that the paper is suitable for publication.

Yours sincerely,

Drs. Honório and Alto

Reviewer #2:

With great interest, I read the paper of Honório et al. submitted for publication in Viruses. The authors provide data that can help to interpreted the possible role of vertical transmission in epidemiology of arboviruses (here Chikungunya virus) in the field. The fact that they found heterogeneous CHIKV dissemination and vertical transmissions. The work has been executed with great care. However, I would like the authors to address the following, rather editorial issues.

 Answer: The authors thank the reviewer #2 for his/her review and valuable comments. The questions raised by the reviewer helped us to  improve the manuscript.

  Scientific comment: It is not clear to me from the Material and Methods whether the females held individually or in cages (how many). It would be interesting to know whether the variation was due to variation between females (e.g. did for example only one female infected eggs). Please elaborate.

Answer: During the first and the second gonotrophic cycle Ae. albopictus females were kept in cages (50 per cage). After the delivery of the non-infectious bloodmeal, females were cold-anesthetized, and fully engorged mosquitos were placed individually in tubes.

 This information has now been added to the Materials and Methods section.

Editorial comments:

 Page 1, line 18: Please add “mosquito” before infected parent as it is unclear during Reading whether you mean the vector or humans.

 Answer: We have corrected this issue.

 Page 3 line 92, spacing between 0.3m3 is missing

 Answer: We have corrected this issue.

 Page 3 line 93: omit extra “t” from mating

 Answer: We have deleted the extra “t”.

 Page 4 Figure 1: I would redraw this diagram as it rather confusing as the text boxes do not contain comparable content. It is not clarifying what is done and what is tested. Further;

 I would add that the original Aedes albopictus were F2 from mosquito eggs collected from the two countries.

Why are there circles around the second mosquito in the diagram? Because legs were tested? Confusing.

Since you did not test virus in eggs, but in adults raised from eggs from infected females, I would rephrase the text to the second and latest batch eggs in the diagram.

 Answer: We would like to thank the reviewer for bringing to our attention this information about Figure 1. We corrected these issues and added information that the original Aedes albopictus were F2 from mosquito eggs collected from the two countries. Also, we clarified information about the second mosquito in the diagram, separating legs from the body and including information about tested legs. We rephrased the text to the second and latest batch eggs in the diagram to adults raised from infected females’ eggs in the second gonotrophic cycle were tested for CHIKV.

 Page 5: line 173-175. The numbers in the text do not agree with those shown in the table. In addition, the Table 1 is confusing to me. Please check the headings of the columns.

Further Total and mean of what (pools?). Minimum-maximum of what (per female, per pool, per test, per cage???).

 Answer: We would like to thank the reviewer for bringing to our attention that Table 1 was not sufficiently detailed and that the numbers in the text do not agree with those shown in the table. We corrected the numbers in the text to reflect Table 1. Table 1 has been changed, so that descriptive statistics are more presentable to the reader. We added a column to the left of the table, indicating what is the mosquito stage (eggs, pupae, adult female and adult male) that the descriptive statistics are referred to. We also included the following description of how each descriptive statistics was calculated after the table title: “Mosquito stages describe each of the stages that were analyzed. Descriptive statistics refers to the total number of each mosquito stage at the end of the second gonotrophic cycle, the mean (standard deviation) for each mosquito stage and the minimum and maximum observed values for each stage. Adults were separated into two entries, one for females and another for males. Positive/Total adult pools refers to the absolute number of pools that were tested positive for CHIKV vertical transmission infection and the total number of pools that were tested for each mosquito sex. Minimum Infection Rate stands for the number of positive pools / total number of tested mosquitos (stratified by sex) x 1,000, as per CDC recommendations.”

 Page 6: Figure 2. The content of the graph could be easily reflected in a table or joint in figure 3.

 Answer: We thank the reviewer for its valuable suggestion. However we chose two different figures because they relate to different statistical models used with two different response variables (one is to illustrate a GLM, the other to illustrate a one-way ANOVA). Both reflect two independent components that we tested in our experiments, one relating to CHIKV disseminated infection in parental females and the other relating to viral titer, and we believe that showing both figures separately will have a more clear impact for potential readers of the manuscript.

 Page 8: Line 224. It is bit odd to refer to “these authors” when the majority is the same as the authors of this paper. Please rephrase.

 Answer: We have rephrased this section. Specifically, we have changed “These authors” to “Also”.

Round  2

Reviewer 1 Report

The Authors have addressed my concerns.